# SCALABLE AI SAFETY VIA DOUBLY-EFFICIENT DEBATE

## ABSTRACT

The emergence of pre-trained AI systems with powerful capabilities across a diverse and ever-increasing set of complex domains has raised a critical challenge for AI safety as tasks can become too complicated for humans to judge directly. Irving et al. (2018) proposed a debate method in this direction with the goal of pitting the power of such AI models against each other until the (mis)-alignment identification problem is broken down into a manageable subtask. While the promise of this approach is clear, the original framework was based on the assumption that the honest strategy is able to simulate *deterministic* AI systems for an *exponential* number of steps, limiting its applicability. In this paper, we show how to address these challenges by designing a new set of debate protocols where the honest strategy can always succeed using a simulation of a *polynomial* number of steps, whilst being able to verify the alignment of *stochastic* AI systems, even when the dishonest strategy is allowed to use exponentially many simulation steps.

## 1 INTRODUCTION

Large language models (LLMs) have demonstrated emergent capabilities, including the ability to follow natural-language instructions, use various tools, and perform some types of general-purpose abstract reasoning and planning (Saunders et al., 2022; Yao et al., 2023; Menick et al., 2022; Zhou et al., 2023). Thus far, human feedback on LLM outputs has been used to improve the alignment between the behavior of these models and their designer's intent (Ouyang et al., 2022). However, these models are increasingly being used to perform complex tasks that can be viewed as the writing and execution of general-purpose computations described in natural language, where at each step the model is invoked with context given by some set of previous model outputs (Lu et al., 2023). As the complexity of such tasks scales, the ability to provide direct human feedback for training on long complex traces involving reasoning, planning, and taking actions is limited. This limitation leads to the need for new approaches for *scalable oversight* (Leike et al., 2018; Christiano et al., 2018)–where carefully designed protocols involving the interaction of both humans and AI models are used to provide high-quality feedback for training and oversight of complex AI systems.

As a motivating example, consider the case of using a language model to draft a law or a legal contract. Laws and contracts are written in natural language, refer to concepts in the real world, and require human judgement (in the worst case a judge in an actual court) to interpret their meaning. Furthermore, individual passages or even single characters in laws or contracts can have significant real-world consequences, as demonstrated by multimillion-dollar losses suffered by companies and governments due to misplaced commas (Kurtzleben, 2014; BBC, 2017). In order to train a language model to write such high-stakes natural language, it is necessary to be certain that every passage of an extremely long document is correct, where correctness is defined by human judgement. However, requiring human experts to carefully read an entire law or contract produced by a language model to provide the training label for one example is clearly prohibitively expensive. Thus, in this setting it is necessary to design methods for training and oversight that are extremely efficient in their use of human judgements.

A prominent approach to the oversight and safe training of AI systems builds upon the fact that there is a natural high-level correspondence between training techniques in machine learning and interactive proofs in complexity theory, as exemplified by the proposal for AI safety via debate

(Irving et al., 2018). The overall goal of this approach is to enable the design of methods that allow the training of extremely computationally powerful learned models that nonetheless behave as desired, despite only being supervised by much more limited verifiers. For example, while no human Go player can instruct the AlphaZero model (Silver et al., 2017) on what move to make next, the model nonetheless was trained to a super-human level via self-play. This was possible precisely because it is computationally easy to verify which player has won at the end of a game of Go. Using such an approach for training LLMs to produce (and then successfully execute) computations described in natural-language requires some method of scalably verifying that the computations produced actually solve the intended task, and are executed correctly.

The surprising ability of computationally limited verifiers to correctly judge the outputs of much more computationally powerful provers underlies some of the most celebrated results in computational complexity theory. Notably, any polynomial space (and potentially exponential time) computation can be verified by a polynomial time verifier interacting with a computationally unbounded prover i.e. IP=PSPACE (Shamir, 1992). Further, for any problem with solutions which can be verified in polynomial time, one can efficiently encode the solutions in such a way that they can be non-trivially verified by reading only three bits chosen uniformly at random from the encoded solution i.e. the PCP theorem (Arora & Safra, 1998; Arora et al., 1998). Recent work has introduced the notion of doubly-efficient interactive proofs (Goldwasser et al., 2015; Reingold et al., 2021) in the context of delegating computation. Here an untrusted prover is asked to run some polynomial-time computation, and the goal is for a linear-time verifier to interact with the prover in order to accurately judge that the computation was performed correctly. Thus, the time spent by the verifier is much less than the time to run the whole computation.

Unfortunately, all of the methods from the theory of interactive proofs for the highly-efficient verification of computationally powerful provers apply only to tasks with mathematically precise definitions (e.g. find a solution to a problem, given the actual code of an algorithm for verifying that the solution is correct). However, in the case of training a model to follow human intent, the main source of feedback available is black-box access to human judgements of model outputs. Strikingly, when access to a black-box is allowed in computations, the main theorems regarding the power of interactive proofs (e.g. IP=PSPACE and the PCP theorem) are actually false (Chang et al., 1994; Fortnow, 1994). However, the goal of efficient verification of powerful provers with access to black-box judgements can still be achieved by requiring that the provers compete.

We introduce the theoretical model of *doubly-efficient debate*, where two polynomial-time provers compete with each other in order to convince a much more efficient verifier of the correctness of a computation that depends on access to black-box judgements. In this model we prove that, under appropriate assumptions, any polynomial-time computation can be verified using only a constant number of queries to the black-box representing human judgement (and in time linear in the size of a single query). Intuitively, our results show that, for any problem whose solutions can be verified by extremely extensive human reflection, the solutions can also be verified with a constant amount of human judgement and interaction with competing provers. A key requirement, and limitation, for applying our results in real-world settings, is that the debating models must have the ability to produce (potentially extensive) natural-language reasoning traces to solve the problem at hand, in such a way that (potentially extensive) careful human analysis *could have been used* to judge that the reasoning was correct. These theorems open up the door for training models with human feedback via self-play, as even very complex and extensive computations described in natural language can be verified by querying human judgements for only a single step of such a computation.

## 1.1 OUR RESULTS

Our definition of doubly-efficient debate is a complexity-theoretic formalization of a training setup in which two competing AI models attempt to convince a verifier, who has access to human judgements, of the correctness of a solution to a computational problem. At a high-level, the goal is to design protocols where (1) the model arguing for the correct solution convinces the verifier without expending computational effort much greater than would be necessary to correctly solve the problem by itself, and (2) the verifier makes a number of queries to human judgements that does not grow (i.e. is a fixed constant) with respect to the computational effort required to solve the problem. The details of the definition appear in Section 4. Recalling the example of models writing laws or contracts, the above goal would allow for training feedback on an entire legal contract, by showing

only a small, fixed (independently of the contract length) number of sentences to a human rater, allowing for scalable training of such models.

In the subsequent sections we prove theorems achieving this high-level goal in several settings. As a warm-up, in Section 5 we give protocols achieving the goal when human judgements are modeled as deterministic, and the competing models are given explicit natural language instructions to follow. In order to better capture the fuzzy nature of human judgement, we then extend these results to the setting where human judgements are stochastic in Section 6. Finally, in Section 7 we prove theorems achieving our goal in the case where the models are asked to come up with a proposed solution themselves, and then are required to justify the correctness of the solution with a natural-language argument. We also include in the supplementary material a machine-verifiable (in lean) formalization of the proof of the main theorem of Section 6.

## 2 RELATED WORK

The work most closely related to ours is the debate proposal by Irving et al. (2018), which proposed the setup of natural-language debates between AI models judged by humans. The original proposal showed that debates between two provers could naturally capture the complexity class PSPACE. Follow-up work of Barnes & Christiano (2020b) introduced cross-examination, which extends the power of debate to all of NEXP. This prior theoretical work models both provers in the debaters as computationally unbounded, which leaves open the question of the ability of actual models to efficiently implement the protocols, and whether there may be an advantage for the dishonest prover in a computationally bounded setting. Our model of doubly-efficient debate makes progress both of these questions, by giving debate protocols where the honest prover always has a winning strategy implementable in polynomial time, even when the dishonest prover is allowed unbounded computation.

The model of doubly-efficient debate is inspired by doubly-efficient interactive proofs in computational complexity first introduced in Goldwasser et al. (2015). The original purpose of this model was to capture the situation where a verifier wants to delegate a polynomial time computation to an untrusted prover, while spending much less time to verify that the computation was performed correctly. Later Reingold et al. (2021) gave the best results currently know for delegating space-bounded computation. See also Goldreich et al. (2018) for a survey of these results. Other related work connecting interactive proofs and machine learning includes Wäldchen et al. (2022), which uses the model of Merlin-Arthur (MA) proof systems in order to achieve formal interpretability of classifier outputs.

The doubly-efficient debate protocols we design are strongly connected to the idea of *process-based feedback* (Stuhlmüller & jungofthewon, 2022; Uesato et al., 2022), where the goal is to directly supervise the reasoning process of an AI system, rather than just the final outcome. Our protocols can be interpreted as a type of process-based feedback where two AI systems compete to convince a limited verifier that a given outcome has been arrived at by a (possibly complex) reasoning process that the verifier would endorse. On the safety side, there have been various proposals that directly supervise language models with human feedback (Ouyang et al., 2022), as well as with additional data from external sources (Menick et al., 2022). There has also been work that utilizes language models to improve supervision of language models including Constitutional AI (Bai et al., 2022) and self-critique (Saunders et al., 2022). There are also alternatives to debate as approaches to scalable oversight including recursive reward modelling (Leike et al., 2018) and iterated amplification (Christiano et al., 2018). Another line of related work on LLMs that motivates the need for scalable oversight is the design of schemes for prompting language models to perform increasingly complex tasks. Notable examples include Chameleon (Lu et al., 2023), ReAct (Yao et al., 2023), and the direct use of language models as prompt engineers (Zhou et al., 2023).

## 3 PRELIMINARIES

We will use the notation $[n] = \{0, 1, \ldots, n\}$. For a vector $x \in \{0, 1\}^n$ and a subset $I \subseteq [n]$ we write $x_S$ to denote the restriction of $x$ to the set of coordinates $i \in I$. For a real number $p > 0$ and positive integer $d$.

We will model computations as Turing machines $M$ with input $x \in \{0,1\}^n$, that additionally have access to an oracle $\mathcal{O}$, which we refer to as oracle Turing machines. Formally, for $l = l(n)$ an *oracle* is a function $\mathcal{O} : \{0,1\}^l \to \{0,1\}$. An *oracle Turing machine* $M$ is a Turing machine with the additional ability to write a query $z \in \{0,1\}^l$ onto its tape, after which it will receive a response $\mathcal{O}(z)$ in one step. We use the notation $M^{\mathcal{O}}$ to indicate the oracle machine $M$ where the queries $z$ are answered by the oracle $\mathcal{O}$. We will also consider the setting where the oracle $\mathcal{O}$ is stochastic, in which case the response to each oracle query $\mathcal{O}(z)$ is an independent $\{0,1\}$-valued random variable. In the LLM setting, the machine $M$ corresponds to a set of natural language rules and instructions, and the oracle $\mathcal{O}$ represents human judgement along with any other external black-box feedback the model may receive (e.g. results from search-query, observations from a camera or sensor, outputs of API calls).

A *language* $L \subseteq \{0,1\}^*$ is a subset of finite-length strings. A deterministic oracle Turing machine $M$ decides a language $L$ with oracle $\mathcal{O}$ if it holds that $M^{\mathcal{O}}(x) = 1 \iff x \in L$. A probabilistic oracle Turing machine $M$ decides a language $L$ with oracle $\mathcal{O}$ if it holds that $x \in L \implies \mathbb{P}[M^{\mathcal{O}}(x) = 1] > \frac{2}{3}$ and $x \notin L \implies \mathbb{P}[M^{\mathcal{O}}(x) = 1] < \frac{1}{3}$. For LLMs, the language $L$ corresponds to some class of problems describable in natural language, each with a yes or no answer that may depend on human judgement or other black-box feedback encoded by the oracle $\mathcal{O}$. The strings $x \in L$ are the problems where the answer is yes, and $x \notin L$ the problems where the answer is no. As is usual this can be extended to search problems (where the answer is polynomial length) by classical search-to-decision reductions.

**Definition 3.1.** A language $L$ is in $\text{NP}^{\mathcal{O}}$ if there is a polynomial-time oracle machine $M$ such that: $x \in L$ if and only if there exists a witness $w$ of length polynomial in $|x| = n$ such that $M^{\mathcal{O}}(x, w) = 1$.

**Definition 3.2.** A language $L$ is in $\text{MA}^{\mathcal{O}}$ if there is a probabilistic oracle machine $M$ and a polynomial $p(n)$ such that:

- $x \in L \implies \exists w$ of length $p(n)$ s.t. $\mathbb{P}[M^{\mathcal{O}}(x, w) = 1] > \frac{2}{3}$.

- $x \notin L \implies \forall w$ of length $p(n)$, $\mathbb{P}[M^{\mathcal{O}}(x, w) = 1] < \frac{1}{3}$.

For the LLM setting, languages in $\text{NP}^{\mathcal{O}}$ and $\text{MA}^{\mathcal{O}}$ correspond to problems $x$ describable in natural language, where a correct solution (the witness $w$) can be verified by polynomially many human judgements of a potentially polynomial length transcript arguing that $w$ is a solution to $x$. These sorts of problems are arguably the most important for safety and scalable oversight, as they correspond to the case where the LLM proposes a plan $w$ in natural language, and goes through a potentially quite long sequence of steps to argue that execution of the plan will have the desired outcome.

The protocols establishing the power of debate in terms of standard complexity classes rely on producing verifiable transcripts of some prescribed computation. A *transcript* of a time $T$ computation of machine $M$ on input $x$ is a string $y \in \{0,1\}^T$, where $y_t$ is the bit written at the current head position of $M$ in time step $t$. We will assume that the $T$-th coordinate of the transcript is equal to the output of $M$ on $x$ i.e. $y_T = M(x)$. In the context of LLMs executing polynomial-length computations from natural-language instructions, the transcript is just the string of tokens output by the model. Given a transcript $y$, the subset of coordinates $I_{M,x}(t) \subseteq [T]$ of $y$ *relevant* to coordinate $t \in [T]$ are the coordinates of the transcript that are read by $M$ when computing $y_t$. When the machine $M$ and input $x$ are obvious from context we will write $I(t)$ for the set of relevant coordinates. For standard Turing machines (without access to an oracle), the set of relevant coordinates has size $O(1)$, but for oracle Turing machines may be as large as $l$.

## 4 DEBATE

A *debate* (Irving et al., 2018) is given by a triple $(A, B, V)$ of oracle Turing machines, an oracle $\mathcal{O}$, and a common input $x$ of length $n$. The machines $A$ and $B$ are called *provers* and $V$ is called the *verifier*. A debate consists of $k = k(n)$ rounds, during which the provers exchange messages. In round $i \in [k]$ prover $A$ sends a message $a^{(i)} = A^{\mathcal{O}}(x, a^{(1)}, b^{(1)}, \dots a^{(i-1)}, b^{(i-1)})$ and prover $B$ sends a message $b^{(i)} = B^{\mathcal{O}}(x, a^{(1)}, b^{(1)}, \dots a^{(i-1)}, b^{(i-1)})$ which can be read by all parties involved. We let $\boldsymbol{a} = (a^{(1)}, \dots a^{(k)})$ and $\boldsymbol{b} = (b^{(1)}, \dots b^{(k)})$ denote the full transcript of the messages sent by

each prover. At the end of the $k$-th round, the verifier runs $V^{\mathcal{O}}(x, \boldsymbol{a}, \boldsymbol{b})$ and outputs either zero or one. As defined, the two provers each send a message in one round, but this also captures the case of taking turns by having them alternate sending empty messages.

## 4.1 Doubly-efficient debate

Different variants of debate arise depending on the computational power and/or limitations of the provers and the verifier.

**Definition 4.1.** A $(P_{time}, V_{time}, q)$-*debate protocol* is given by a triple of oracle Turing machines $(A, B, V)$ where $A$ and $B$ run in time $P_{\text{time}}$, and $V$ runs in time $V_{\text{time}}$ and makes $q$ oracle queries. Let $1 \geq c > \frac{1}{2} > s \geq 0$. A debate protocol decides a language $L$ with completeness $c$ and soundness $s$ if:

- **Completeness:** If $x \in L$ then for all (unbounded time) oracle Turing machines $B'$ the debate $(A, B', V)$, with oracle $\mathcal{O}$, and input $x$ satisfies $\mathbb{P}[V^{\mathcal{O}}(x, \boldsymbol{a}, \boldsymbol{b}) = 1] \geq c$.

- **Soundness:** If $x \notin L$ then for all (unbounded time) oracle Turing machines $A'$ the debate $(A', B, V)$, with oracle $\mathcal{O}$, and input $x$ satisfies $\mathbb{P}[V^{\mathcal{O}}(x, \boldsymbol{a}, \boldsymbol{b}) = 1] \leq s$.

When $c = 1$ and $s = 0$ we say that the debate protocol *deterministically decides* $L$.

For deterministic oracle machines, as there is no randomness, it will always be the case that $c = 1$ and $s = 0$ i.e. that the honest prover always wins. For stochastic oracle machines the definition requires a constant gap between $c$ and $s$, which as usual can be amplified by repeating the protocol multiple times and taking the majority outcome. A debate protocol specifies the behavior of honest provers for both of the cases $x \in L$ and $x \notin L$. Additionally, it is required that the honest prover wins the debate with higher probability while running in time $P_{\text{time}}$, against any (computationally unbounded) strategy by the dishonest prover. Note that this requirement gives a complexity theoretic formalization of the intuitively desirable property that debates should be structured so that it is easier to tell the truth than to lie.

The original definition of debate requires $V_{\text{time}}$ to be polynomial in $n$, but allows $P_{\text{time}}$ to be unbounded. *Doubly-efficient debate* refers to the setting where $P_{\text{time}}$ is polynomial in $n$ and $V_{\text{time}}$ is linear in $l$, the length of each oracle query, and linear (or ideally sub-linear) in $n$, and $V$ makes a sub-linear (or ideally constant) number $q$ of oracle queries.

## 4.2 Training and inference with debate

We first clarify the relationship between our formal theoretical model of debate, and practical training setups.

**The oracle $\mathcal{O}$:** Our theoretical model gives both the provers and the verifier access to an oracle $\mathcal{O}$ representing human judgement. The prover access to the oracle corresponds to the fact that the powerful pre-trained models already have extensive knowledge of human judgement on many questions, and will gain more knowledge throughout the training process. The verifier access to the oracle $\mathcal{O}$ corresponds to actual judgements by human raters.

**At training time:** The pretrained models $A$ and $B$ are trained via self-play to win the zero-sum game given by the debate protocol. When a model queries $\mathcal{O}$ this corresponds to using either pre-trained knowledge, or knowledge gained during self-play, to predict the answer that a human would give to the oracle query. When the verifer $V$ queries $\mathcal{O}$ this corresponds to asking an actual human rater to judge an oracle query. The distinction between pre-trained knowledge for provers, versus queries to human judgements for the verifier is critical, because it means that the training cost in terms of number of queries to human judgement is equal to the number of verifier queries to $\mathcal{O}$. Thus, as long as the number of verifier oracle queries is bounded, the training procedure can scale to arbitrarily complex computations by the models $A$ and $B$, while still only requiring a bounded number of human judgements.

**At inference time:** When a model is asked to solve a problem or follow complex natural language instructions at inference time, the debate protocol is still run. However, no human feedback is used. Instead the output of the model $A$ is trusted, as long as $B$ does not abort/point out a flaw.

## 5 DETERMINISTIC DEBATE

Doubly-efficient debate can decide any problem solvable in bounded space with verifier time that is nearly-linear in the space used, and only a constant number of verifier queries to $\mathcal{O}$.

**Theorem 5.1.** *Let $L$ be any language decidable by an oracle Turing machine $M$ in time $T = T(n)$ using space $S = S(n)$. Then there is a $(O(T \log T), O(S \log T), O(1))$-debate protocol deterministically deciding $L$.*

The proof appears in Section B. One can compare Theorem 5.1 to the setting of doubly-efficient interactive proofs where there is a single prover (and without any black-box oracles). Reingold et al. (2021) show that any time $T$ space $S$ computation can be decided by a doubly-efficient interactive proof in time $O(S^2 \operatorname{polylog} T)$. It is currently an open question whether this can be improved to $O(S \operatorname{polylog} T)$ (Goldreich et al., 2018). Additionally, the protocol of Reingold et al. (2021) is quite complex, and relies on prior work in interactive proofs including the PCP theorem, so does not apply in the presence of a black-box oracle.

The protocol achieving Theorem 5.1 is given in Figure 3 in Section A. The basic idea (which has been used in many classical PSPACE-completeness results), is to have $A$ output a supposed middle configuration of the computation of $M(x)$. Then $B$ decides to recursively call the protocol on either the first or the second half of the computation. This recursion bottoms-out at a single transition of the machine $M$ which can be checked by $V$.

### 5.1 CROSS-EXAMINATION

The power of debate can be increased by allowing for cross-examination, where multiple copies of each debater are questioned independently. Intuitively this should give more power, as the independent copies must give consistent answers to the queries asked, and so may have more difficulty lying.

**Definition 5.2.** A debate with *cross-examination* is a debate where $A, B$, and $V$ can query independent, non-communicating copies of both $A$ and $B$. Furthermore, the verifier is not required to read the entire transcript of the debate, but can selectively query a subset of the transcript. A debate protocol with cross-examination is a debate protocol where the debates appearing in the completeness and soundness case allow cross-examination.

The definition of cross-examination is quite natural when considering language-model debaters. In this case, the ability to query independent copies can be achieved by either running multiple copies of the same LLM, or more efficiently by simply querying the same LLM with any previous messages in the debate removed from the context. Our next theorem shows that doubly-efficient debate with cross-examination can decide any problem solvable in polynomial time, using only $O(l)$ verifier time (and hence only $O(1)$ oracle queries).

**Theorem 5.3.** *Let $L$ be any language decidable by an oracle Turing machine $M$ in time $T = T(n)$ with oracle queries of length $l$. Then there is a $(O(T \log T), O(l \log T), O(1))$-debate protocol with cross-examination deterministically deciding $L$.*

The proof appears in Section B. The protocol achieving Theorem 5.3 is given in Figure 4 in Section A. Cross-examination allows for a simple and powerful protocol where $A$ outputs the whole transcript of the computation $M(x)$, $B$ outputs the location of a supposed mistake by $A$, and $V$ checks only this location.

## 6 STOCHASTIC DEBATE

In this section we give a debate protocol for any language $L$ decidable by a probabilistic oracle machine $M$ with access to a stochastic oracle $\mathcal{O}$. In the LLM setting, the oracle $\mathcal{O}$ is intended to model human judgement, as well as other types of responses from nature (e.g. real world data or observations). Thus, the oracle $\mathcal{O}$ must be stochastic in order for the model to be relevant in most real-world scenarios. However, access to a stochastic oracle introduces an additional subtlety, where changes on the order of $O(\frac{1}{T})$ in the oracle's distribution may add up to an $O(1)$ difference in the final output probability over the course of a time $T$ computation. To account for this issue, we require an additional Lipschitzness assumption for the machine $M$.

---

**Debate protocol for a stochastic oracle**

---

All parties have access to an $\mathcal{O}$, input $x \in \{0, 1\}^n$, and $K$-Lipschitz probabilistic oracle machine $M$.

$A$ claims that $\mathbb{P}[M(x) = 1] \geq \frac{2}{3}$, and $B$ disputes this claim.

1. Let $y$ be the random variable representing the transcript of $M$ on input $x$, and let $d = \lceil 150K \rceil$.

2. The debate proceeds in a series of $T$ rounds, in lexicographic order for $t \in [T]$. In round $t$:

   (a) $A$ outputs a probability $\hat{p}_t \in [0, 1]$, which is supposed to equal $p_t = \mathbb{P}[y_t = 1 \mid y_{I(t)} = a_{I(t)}]$.

   (b) $A$ queries an independent copy of $B$ for a real number $z_t^A \in [0, 1]$, and $B$ queries an independent copy of $A$ for a real number $z_t^B \in [0, 1]$. Both $z_t^A$ and $z_t^B$ are supposed to be sampled uniformly at random from $[0, 1]$.

   (c) Let $z_t = z_t^A + z_t^B \mod 1$.

   (d) $A$ sets $a_t = 1$ if $z_t \leq \hat{p}_t$, and $a_t = 0$ otherwise.

   (e) At this point $B$ can abort the protocol in round $t$, and is supposed to do so if $A$ has not followed the protocol.

3. If the protocol was aborted at round $t$:

   (a) The verifier $V$ draws $r = 192d^2 \log 100$ samples of $y_t$ conditioned on $y_{I(t)} = a_{I(t)}$ to obtain an estimate $\hat{p}_t^{\mathcal{O}}$ of the probability $\mathbb{P}[y_t = 1 \mid y_{I(t)} = a_{I(t)}]$ via the sample mean. Note that if $y_t$ is supposed to be the output of an oracle query this can be done with $r$ queries to $\mathcal{O}(a_{I(t)})$, otherwise $y_t$ is a deterministic function of $a_{I(t)}$ given by one step of $M$.

   (b) $V$ checks if $\left| \hat{p}_t^{\mathcal{O}} - \hat{p}_t \right| \geq \frac{3}{8d}$ and outputs 0 if so and 1 otherwise.

4. If the protocol was not aborted, then $V$ outputs $a_T$.

---

Figure 1: Doubly-efficient debate protocol for a stochastic oracle.

**Definition 6.1.** For $K > 0$, a probabilistic oracle machine $M$ is $K$-Lipschitz at oracle $\mathcal{O}$ if, for any other oracle $\mathcal{O}'$,

$$\sup_x \left| \mathbb{P}[M^{\mathcal{O}}(x) = 1] - \mathbb{P}[M^{\mathcal{O}'}(x) = 1] \right| < K \sup_z \left| \mathbb{P}[\mathcal{O}(z) = 1] - \mathbb{P}[\mathcal{O}'(z) = 1] \right|$$

In other words, if $M$ is run with any oracle which assigns similar probabilities to $\mathcal{O}$, the probability that $M$ outputs 1 should change by at most a $K$ factor more than the maximum differences in the oracle probabilities. Observe that every time-$T$ stochastic oracle machine is $K$-Lipschitz for $K = O(T)$.

**Theorem 6.2.** *For $K > 0$, let $L$ be any language decidable by a $K$-Lipschitz probabilistic oracle Turing machine $M$ in time $T = T(n)$ with oracle queries of length $l$. Then there is a $(O(K^2 T \log T), O(K^2 + l \log T), O(K^2))$-debate protocol with cross-examination deciding $L$ with completeness $\frac{3}{5}$ and soundness $\frac{2}{5}$.*

The proof appears in Section D. The debate protocol promised in Theorem 6.2 is given in Figure 1. As usual the protocol describes the prescribed behavior of the honest provers, but emphasizes that dishonest behavior may occur. The protocol proceeds in $T$ rounds, where in each round $A$ proposes a probability distribution over the next bit given the computation so far. Then $A$ and $B$ use cross-examination to engage in a coin-flipping protocol (Steps 2.b. and 2.c.) in order to sample the next bit of the computation from the distribution proposed by $A$. Finally, $B$ can abort the protocol at any round $t$, whereupon $V$ samples from $\mathcal{O}$ in order to check if $A$'s proposed distribution at round $t$ is correct.

Theorem 6.2 delivers non-trivial savings in verifier time and query complexity whenever $K = o(\sqrt{T})$. In particular, the most interesting case occurs for $K = O(1)$ i.e. when $K$ is a constant independent of $T$.

**An Example for Theorem 6.2.** Suppose a researcher wants to perform a meta-analysis of the existing literature on whether caffeine improves learning in college students. The researcher could give a high-level plan in natural language for how to accomplish this task:

1. Search the literature for studies on caffeine consumption and learning in college students.
2. Among the studies found, select those studies which meet the following quality criteria:
    (a) Do not include any studies that contain clear methodological flaws.
    (b) Only include randomized control trials which measure final exam performance after at least six weeks of intervention.
3. Perform the actual statistical analysis using a random effects model.
4. Measure sources of heterogeneity between the included studies using meta-regression.

The above natural language plan is the machine $M$ in Theorem 6.2. Instructions such as 2.(a) correspond to queries to the human judgement oracle $\mathcal{O}$, as whether a study contains a clear flaw is a question of judgement. To run the debate protocol the model $A$ is prompted to carry out the plan. $A$ will then carry out the instructions above while carefully listing everything relevant to show that the instructions are being followed correctly e.g. text from each study to explain why it is included. At the same time $B$ will examine each step made by $A$ and possibly abort if $B$ believes the step has been made incorrectly.

To see how the protocol in Figure 4 works in this case suppose that $B$ aborts when $A$ decides not to include a particular study at Step 2.(a). The protocol requires $A$ to output a probability estimate for how likely a human is to say that the study contains a clear methodological flaw. Assume that $A$ has output probability $\frac{3}{4}$ that the study contains a flaw. Then when $B$ aborts, the question of whether the study has the flaw claimed by $A$ will be sent to human raters. If the fraction of raters who agree with $A$ about the flaw is sufficiently close to $\frac{3}{4}$ then $A$ wins. Otherwise $B$ wins.

## 7 DOUBLY-EFFICIENT DEBATE WITH A WITNESS

One should also consider the situation where the debaters additionally are able to non-deterministically produce a polynomial-sized witness $w$ for the membership of $x$ in an $\mathrm{NP}^{\mathcal{O}}$ or $\mathrm{MA}^{\mathcal{O}}$ language $L$. In this case the honest debaters are required to run in polynomial time as in Definition 4.1 when additionally given the witness $w$ as input. This case corresponds to the setting where an LLM proposes some solution to a very complex problem, and then argues for the correctness of the solution via a polynomially long natural-language argument. Our results in this section prove that, as long as this argument can be verified via extensive human reflection, then there is a debate protocol that allows a human judge to only check a constant number of steps of the argument when interacting with two competing models. The protocols of Figure 4 and Figure 1 then carry over immediately where the machine $M$ is the polynomial-time verifier for $L$ and both $x$ and the witness $w$ are given as input.

---

**Debate protocol with a witness for time $T$**

All parties have access to an oracle $\mathcal{O}$, input $x \in \{0, 1\}^n$ and the code of a time $T$ oracle machine $M$ for verifying witnesses for a language $L$.

$A$ claims that $x$ is in $L$, and $B$ disputes this claim.

1. $A$ outputs a claimed witness $w$ for the membership of $x$ in $L$.
2. If the oracle $\mathcal{O}$ is deterministic, run the protocol of Figure 4 with input $(x, w)$ and machine $M$.
3. If the oracle $\mathcal{O}$ is stochastic, run the protocol of Figure 1 with input $(x, w)$ and machine $M$.

---

Figure 2: Doubly-efficient debate protocol with a witness.

The protocol given in Figure 2 leads immediately to the following theorems.

**Theorem 7.1.** *Let $L \in \mathrm{NP}^{\mathcal{O}}$ and let $M$ be the time $T = T(n)$ verifier for $L$ with oracle queries of length $l$. Then there is a $(O(T), O(l \log T), O(1))$-debate protocol with cross-examination deterministically deciding $L$ given a witness $w$.*

**Theorem 7.2.** *For $K > 0$, let $L \in \mathrm{MA}^{\mathcal{O}}$ be a language with a verifier $M$ that is a $K$-Lipschitz, time $T = T(n)$ probabilistic oracle machine, with oracle queries of length $l$. Then there is a $(O(K^2 T \log T), O(K^2 + l \log T), O(K^2))$-debate protocol with cross-examination deciding $L$ with completeness $\frac{3}{5}$ and soundness $\frac{2}{5}$, given a witness $w$.*

The proofs of Theorem 7.1 and Theorem 7.2 appear in Section E.

**An Example for Theorem 7.2.**  The simplest example for Debate with a witness is to take the setting of the example given for Theorem 6.2 above, but drop the assumption that the high-level natural language plan is given in advance. Instead one could assume that the researcher just asks the question "Does caffeine improve learning in college students? Produce a high-level plan to answer this question, give a step-by-step justification of the correctness of the plan, and then execute the plan." Here the high-level plan plus the justification for it's correctness is the witness $w$ produced by $A$ in Theorem 7.2. The machine $M$ is given by the natural-language instructions (implicit in the prompt above):

1. Go through every step of the plan and the corresponding justification and check that it is clearly correct.
2. Execute the plan.

Clearly many calls to the human-judgement oracle $\mathcal{O}$ are necessary to execute $M$ with witness $w$. The protocol at this point will proceed as in the example above, with $A$ going step-by-step through $M$, and $B$ aborting for a claimed incorrect step.

## 8    Conclusion and Open Problems

We introduced the theoretical model of doubly-efficient debate, where two polynomial-time provers compete to convince a significantly more efficient verifier that they have correctly solved a computational problem that depends on black-box access to human judgements. At a high level, our main theorems show that doubly-efficient debate can be used to allow for the verification of arbitrary polynomial time computations using only a constant amount of human judgement. The overall aim of this model is to provide theoretical grounding for scalable oversight of powerful AI systems, using limited human feedback, and our theorems represents an initial step towards this goal. However, many open questions remain.

**The Power of the Provers:**   The theorems in this paper apply to the setting of verifying computations that *could have been* verified by a human reading the entire (polynomial-length) transcript of the computation. How can the theoretical model be extended to settings where this is not possible? On the one hand, our model assumes the AI systems implementing the provers are powerful enough to very accurately simulate human judgements on any query. This may attribute too much power to these systems. Is it possible to relax the accuracy requirements for the provers e.g. by giving the provers access to an approximately correct oracle $\mathcal{O}'$?

On the other hand, extremely powerful AI systems may be able to perform computations that, while polynomial time, do not have any polynomial length human-verifiable transcript. The original debate proposal with unbounded provers captures all of PSPACE, and thus is able to efficiently interrogate implicitly-represented exponential length transcripts. However, allowing both provers in the theoretical model to be unbounded runs into what is referred to by Barnes & Christiano (2020a) as the *obfuscated argument problem*, where a dishonest prover can in polynomial time produce an argument that would require the honest prover exponential time to refute. Is there some intermediate model where the honest prover always has an efficient strategy, but the computation to be verified does not require a polynomial-length human-verifiable transcript?

**The Power of the Verifier:**   Human judgement is fallible in many ways. Furthermore, current approaches to scalable oversight, such as reinforcement learning from human feedback, generally train AI models (known as reward models) to approximate human judgements from a limited number of samples. Thus, in the practical settings of interest the oracle $\mathcal{O}$ used by the verifier is likely to be flawed. Theorem 6.2 partially addresses this problem by making each response of $\mathcal{O}$ stochastic, and allowing for the verification of any computation that outputs the correct answer with a constant advantage over random guessing. Is it possible to extend these results to settings where $\mathcal{O}$ gives incorrect answers on some subset of queries? There are many possible models in this direction e.g. is there a class of computations that can be verified by debate, where the oracle may make errors on an arbitrary subset of limited size? Alternately, can debate verify computations where the oracle makes arbitrary errors on a randomly selected subset of queries?

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

# A  DETERMINISTIC DEBATE PROTOCOLS

In this section we include the full specifications of the protocols for deterministic debate.

---

**Debate protocol for time $T$ and space $S$**

All parties have access to an oracle $\mathcal{O}$, input $x \in \{0,1\}^n$, and the code of the time $T$ space $S$ oracle machine $M$.

$A$ claims that $M(x) = 1$, and $B$ disputes this claim.

1. The debate proceeds recursively in a series of $O(\log T)$ rounds. Let $z_0 = x$ and $t_0 = T$. The $k$-th round begins with $A$ arguing that the execution of $M$ starting in configuration $z_k$ ends in configuration $z'_k$ in $t_k$ steps.

   (a) $A$ outputs the configuration $a_k$, which is supposed to be equal to the middle configuration of $M$ after $\frac{t_k}{2}$ steps starting from $z_k$.

   (b) $B$ outputs a bit $b_k$, which is supposed to be 1 if $A$ is lying about the execution of $M$ from $z_k$ to $a_k$, and zero if $A$ is lying about the execution from $a_k$ to $z'_k$.

   (c) If $b_k = 1$ then $A$ enters the next round with $z_{k+1} = a_k$ and $z'_{k+1} = z'_k$. If $b_k = 0$ then $A$ enters the next round with $z_{k+1} = z_k$ and $z'_{k+1} = a_k$. In either case $t_{k+1} = \frac{t_k}{2}$.

2. The verifier $V$ checks that each configuration $a_k$ output by $A$ is a valid configuration of $M$, and that the final two configurations output by $A$ are a valid execution of one step of $M$.

---

Figure 3: Doubly-efficient debate protocol for time $T$ and space $S$.

---

**Debate protocol with cross-examination for time $T$**

All parties have access to an oracle $\mathcal{O}$, input $x \in \{0,1\}^n$ and the code of the time $T$ oracle machine $M$.

$A$ claims that $M(x) = 1$, and $B$ disputes this claim.

1. $A$ outputs a string $a$, which is supposed to be the transcript $y$ of $M$ on input $x$

2. $B$ outputs a location $t \in [T]$ as well as the relevant coordinates $I(t)$, where $A$ has supposedly computed $a_t$ incorrectly.

3. The verifier $V$ reads the relevant bits $a_{I(t)}$ and checks that $a_t$ is correct for the execution of $M$ given these bits. If so $V$ outputs 1, if not 0.

---

Figure 4: Doubly-efficient debate protocol with cross-examination for time $T$.

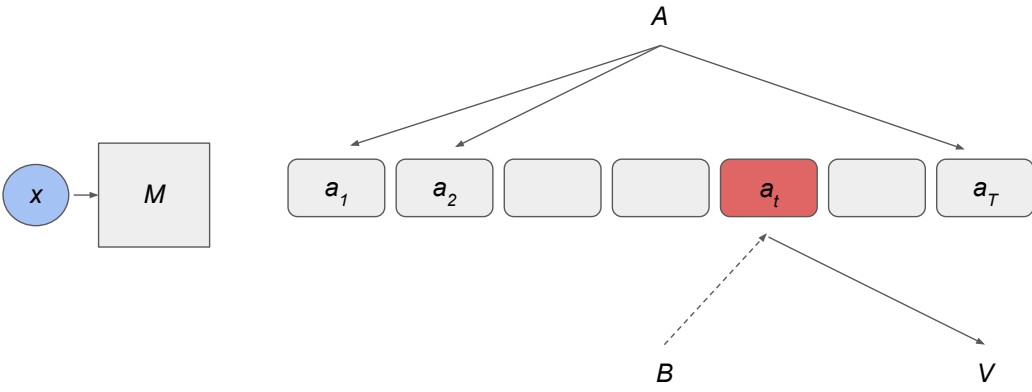

Figure 5: A schematic of the debate protocol with cross examination. The prover $A$ simulates the execution of the machine $M$ on input $x$. The prover $B$ points to a location of an incorrect step $a_t$, and $V$ checks that step.

## B    Missing Proofs for Deterministic Debate

In this section we give the missing proofs for the theorems on the power of deterministic debate protocols, both with and without cross-examination.

*Proof of Theorem 5.1.*

**Completeness**   If $x \in L$ then an honest prover $A$ can run $M(x)$ once to get the transcript $y$ and always output the appropriate middle configuration in Step 1a, no matter which bits $b_k$ are chosen. The messages $a_k$ will then pass all checks in Step 2.

**Soundness**   Suppose $x \notin L$. Then inductively in the $k$-th round a dishonest prover $A'$ must output a middle configuration $a_k$ for which either the first half ($z_k$ to $a_k$) or the second half ($a_k$ to $z_k'$) is not a correct execution of $M$. Then an honest prover $B$ can run $M$ for $\frac{t_k}{2}$ steps from $z_k$ and from $a_k$, and output the appropriate bit $b_k$ indicating which half was incorrect. Thus, in Step 3 either the last two configurations are not a correct step of $M$ or one of the configurations output by $M$ is invalid, so $V$ will reject.

**Efficiency**   The honest prover $A$ only needs to run $M(x)$ once. The honest prover $B$ also only needs to run $M$ for $O(T)$ total steps. The current head position of $M$ can be encoded in $O(\log T)$ bits, so simulating one step of $M$ requires at most $O(\log(T))$ time. The verifier $V$ checks $O(\log T)$ configurations each of size $S$, and simulates one step of $M$ (possibly issuing one query to $\mathcal{O}$), for a total time of $O(S \log T)$.   □

*Proof.*   Proof of Theorem 5.3

**Completeness**   If $x \in L$ the honest prover $A$ can just output the true transcript of $M$ on input $x$, and will pass the test in Step 3 no matter which location $t$ is checked by $V$.

**Soundness**   If $x \notin L$ then a dishonest prover $A'$ must output a transcript $A$ that does not correspond to a correct execution of $M(x)$. In particular, there must be at least one location $t$ where $a_t$ is not correct given the relevant bits $a_{I(t)}$. An honest prover $B$ can then execute $M(x)$ and find such a location, which will in turn cause $V$ to reject in Step 3.

**Efficiency**   The current head position of $M$ can be encoded in $O(\log T)$ bits, so simulating one step of $M$ requires at most $O(\log(T))$ time. Both provers need only simulate $M(x)$ once which takes $O(T \log T)$ time. The verifier only needs to read the at most $O(l)$ relevant bits $a_{I(t)}$, the locations of which can each be encoded in $O(\log T)$ bits, and execute one step of $M$ (possibly making one query to $\mathcal{O}$), which takes a total of $O(l \log T)$ time.   □

## C    The debate game

The verifier $V$ in a debate protocol deciding a language $L$ naturally defines a family of (potentially stochastic) two-player zero-sum games $G(V, x)$, one for each input $x$. The game $G(V, x)$ is defined as follows:

- The strategies available to the first player are all oracle Turing machines $A$, and to the second player all oracle Turing machines $B$.
- The expected payoff to the first player is $\mathbb{P}[V(x, \boldsymbol{a}, \boldsymbol{b}) = 1]$.
- The expected payoff for the second player is $1 - \mathbb{P}[V(x, \boldsymbol{a}, \boldsymbol{b}) = 1]$.

The existence of a $(P_{\text{time}}, V_{\text{time}})$-debate-protocol deciding a language $L$ then has an equivalent statement in game-theoretic language. In particular if $x \in L$ then there is a strategy $A$ for the first player in $G(x, V)$ achieving value at least $c$, regardless of the second player's strategy. Furthermore, the strategy $A$ is a Time $P_{\text{time}}$ oracle Turing machine. Similarly, if $x \notin L$ then there is a strategy

$B$ for the second player in $G(x, V)$ achieving value at least $1 - s$, where $B$ is a time-$P_{\text{time}}$ oracle Turing machine. This equivalent game-theoretic statement gives a justification for the safety of training a model to decide a language $L$ via self-play in the games $G_x$. In particular, the existence of a $(P_{\text{time}}, V_{\text{time}})$-debate protocol means that the prover tasked with arguing for the correct answer always receives a larger expected pay-off, even when restricted to strategies computable in time $P_{\text{time}}$.

## D  MISSING PROOFS FOR STOCHASTIC DEBATE

In this section we give the missing proof for Theorem 6.2 on the power of stochastic debate protocols. For a pair of oracles $\mathcal{O}, \mathcal{O}'$ we will use the notation $\|\mathcal{O} - \mathcal{O}'\|_\infty = \max_z \mathbb{P}[\mathcal{O}(z) = 1] - \mathbb{P}[\mathcal{O}'(z) = 1]$

*Proof.* Proof of Theorem 6.2  We will require the following version of the Chernoff bound throughout the proof: Let $X_1, \ldots, X_N$ be independent Bernoulli random variables, each taking the value 1 with probability $p$ and 0 with probability $(1 - p)$. Let $\hat{\mu} = \frac{1}{N} \sum_{i=1}^N X_i$ be the empirical mean of the random variables. Then,

$$\mathbb{P}\left[|\hat{\mu} - p| \geq s\right] < 2e^{-\frac{s^2 N}{3}}.$$

Since $M$ is $K$-Lipschitz at $\mathcal{O}$, for any oracle $\mathcal{O}'$ satisfying $\|\mathcal{O} - \mathcal{O}'\| < \frac{1}{d}$

$$\left|\mathbb{P}\left[M^{\mathcal{O}'}(x) = 1\right] - \mathbb{P}\left[M^{\mathcal{O}}(x) = 1\right]\right| < \frac{K}{d} \leq \frac{1}{150}. \tag{1}$$

**Completeness**  If $x \in L$, we first describe how the prover $A$ can efficiently follow the prescribed protocol. In each round $t$ the prover $A$ draws $R = 192d^2 \log 100T$ samples from $y_t$ conditioned on $y_{I(t)} = a_{I(t)}$ (this may involve querying $\mathcal{O}(a_{I(t)})$ up to $R$ times). Next $A$ outputs the sample mean $\hat{p}_t$ of these $R$ samples as the probability in step 2.a. Finally, when $A$ is queried for a random integer in step 2.b, $A$ outputs a number $z_t^A \in \{0, \ldots, d\}$ sampled uniformly at random.

Next we analyze the probability that the verifier $V$ accepts when $A$ follows the protocol as described above, and $B'$ is an arbitrary strategy. Let $p_t = \mathbb{P}[y_t = 1 \mid y_{I(t)} = a_{I(t)}]$ be the true probability that $y_t = 1$ conditioned on the execution so far. Let $E_t$ be the event that $|\hat{p}_t - p_t| < \frac{1}{8d}$. Let $H_t$ be the history of all messages sent in the protocol up until the end of round $t$. Let $a(H_t)$ denote the bits $a_1 \ldots a_t$ output in the history $H_t$. We will call a history $H_t$ "good" if $E_{t'}$ occurs and $B'$ does not abort in round $t'$ for all $t' \leq t$ in the history. Let $K_t$ be the event that $B'$ aborts in round $t$.

For the analysis it will be useful to define an alternative oracle machine $M'$. The machine $M'$ is exactly the same as $M$ except that in the final step $T$, if $M$ outputs 1, then with probability $\frac{1}{50}$ $M'$ outputs 0, otherwise $M'$ outputs the same value that $M$ outputs. This implies that, given any initial setting of the transcript $y_{\leq t} = a_{\leq t}$ and any oracle $\mathcal{O}'$,

$$\mathbb{P}\left[M'^{\mathcal{O}'} \to 1 | y_{\leq t} = a_{\leq t}\right] = \frac{49}{50} \mathbb{P}\left[M^{\mathcal{O}'} \to 1 | y_{\leq t} = a_{\leq t}\right] \leq \frac{49}{50}. \tag{2}$$

The proof proceeds by induction for decreasing values of $t \leq T$. The inductive hypothesis is: For any good history $H_t$, there exists an oracle $\mathcal{O}_t$ with $\|\mathcal{O}_t - \mathcal{O}\|_\infty < \frac{1}{d}$ satisfying

$$\mathbb{P}[V \to 1 | H_t] \geq \left(\mathbb{P}\left[M'^{\mathcal{O}_t} \to 1 | y_{\leq t} = a(H_t)\right]\right) \left(1 - \frac{1}{50T}\right)^{T-t}. \tag{3}$$

The base case $t = T$ follows from the fact that, given a good history $H_T$, $V$ simply outputs $a_T$. Thus, if $a_T = 1$ then $V$ outputs 1 with probability one, and $M'$ outputs 1 with probability $\frac{49}{50}$. If $a_T = 0$ both $V$ and $M'$ output 0.

For the inductive case, since $A$ draws $R = 192d^2 \log 100T$ independent samples conditioned on the value of $a_{I(t)}$ to estimate $\hat{p}_t$, the Chernoff bound implies that for any history $H_{t-1}$

$$\mathbb{P}\left[E_t \Big| H_{t-1}\right] \geq 1 - \mathbb{P}\left[|\hat{p}_t - p_t| \geq \frac{1}{8d} \Big| H_{t-1}\right] > 1 - 2e^{-\frac{R}{192d^2}} = 1 - \frac{1}{50T}. \tag{4}$$

Next if $E_t$ occurs and $B'$ aborts, $V$'s decision depends only on the value of $\hat{p}_t$. Thus for any bit $\alpha_t$, the probability that $V$ outputs 1 after taking $r = 192d^2 \log 100$ samples is, by the Chernoff bound,

$$\mathbb{P}[V \to 1 | H_{t-1}, E_t, a_t = \alpha_t, K_t] \geq 1 - \mathbb{P}\left[ \left| \hat{p}_t^{\mathcal{O}} - p_t \right| \geq \frac{1}{4d} \middle| H_{t-1}, E_t, a_t = \alpha_t, K_t \right]$$

$$\geq 1 - 2e^{-\frac{r}{48d^2}} > \frac{49}{50}. \tag{5}$$

Next let $H_t^1$ be the extension of $H_{t-1}$ where $E_t$ occurs, $a_t = 1$ and $B'$ does not abort. Similarly let $H_t^0$ be the extension of $H_{t-1}$ where $E_t$ occurs, $a_t = 0$ and $B'$ does not abort. If $E_t$ occurs, since $A$ samples $z_t^A$ independently of everything else in the protocol, the value of $z_t = z_t^A + z_t^B \pmod 1$ in step 2.c will be uniformly random in $[0, 1]$. Thus, $a_t$ will be set to 1 with probability exactly $\hat{p}_t$ in step $t$ i.e. $\mathbb{P}[a_t = 1 | H_{t-1}, E_t] = \hat{p}_t$. Therefore, using (5) we have

$$\mathbb{P}\left[V \to 1 | H_{t-1}, E_t\right] \geq \mathbb{P}\left[V \to 1 | H_t^1\right] \hat{p}_t \mathbb{P}[\overline{K}_t | H_{t-1}, E_t, a_t = 1]$$

$$+ \frac{49}{50} \hat{p}_t \mathbb{P}[K_t | H_{t-1}, E_t, a_t = 1]$$

$$+ \mathbb{P}\left[V \to 1 | H_t^0\right] (1 - \hat{p}_t) \mathbb{P}[\overline{K}_t | H_{t-1}, E_t, a_t = 0]$$

$$+ \frac{49}{50} (1 - \hat{p}_t) \mathbb{P}[K_t | H_{t-1}, E_t, a_t = 0]$$

$$\geq \min \left\{ \mathbb{P}\left[V \to 1 | H_t^1\right], \frac{49}{50} \right\} \hat{p}_t$$

$$+ \min \left\{ \mathbb{P}\left[V \to 1 | H_t^0\right], \frac{49}{50} \right\} (1 - \hat{p}_t)$$

For a good history $H_t$, let $\mathcal{O}_t$ be the oracle guaranteed to exist by the inductive hypothesis, and define $\mathcal{O}_{t-1}$ to be identical to $\mathcal{O}_t$, except that $\mathbb{P}[\mathcal{O}_{t-1}(a_{I(t)}) = 1] = \hat{p}_t$. Observe that, for any good history $H_t$, the occurence of $E_t$ implies that the oracle $\mathcal{O}_{t-1}$ will satisfy $\|\mathcal{O}_{t-1} - \mathcal{O}\|_\infty < \frac{1}{d}$. Applying the inductive hypothesis (3) followed by (2) yields

$$\mathbb{P}\left[V \to 1 | H_{t-1}, E_t\right] \geq \min \left\{ \left( \mathbb{P}\left[M'^{\mathcal{O}_t} \to 1 | y_{\leq t} = a(H_t^1)\right] \right) \left(1 - \frac{1}{50T}\right)^{T-t}, \frac{49}{50} \right\} \hat{p}_t$$

$$+ \min \left\{ \left( \mathbb{P}\left[M'^{\mathcal{O}_t} \to 1 | y_{\leq t} = a(H_t^0)\right] \right) \left(1 - \frac{1}{50T}\right)^{T-t}, \frac{49}{50} \right\} (1 - \hat{p}_t)$$

$$\geq \left( \mathbb{P}\left[M'^{\mathcal{O}_t} \to 1 | y_{\leq t} = a(H_t^1)\right] \right) \left(1 - \frac{t}{50T}\right)^{T-t} \hat{p}_t$$

$$+ \left( \mathbb{P}\left[M'^{\mathcal{O}_t} \to 1 | y_{\leq t} = a(H_t^0)\right] \right) \left(1 - \frac{1}{50T}\right)^{T-t} (1 - \hat{p}_t)$$

$$= \left( \mathbb{P}\left[M'^{\mathcal{O}_{t-1}} \to 1 | y_{\leq t-1} = a(H_{t-1})\right] \right) \left(1 - \frac{1}{50T}\right)^{T-t}.$$

Therefore, combining the above calculation with (4) yields

$$\mathbb{P}\left[V \to 1 | H_{t-1}\right] = \mathbb{P}\left[V \to 1 | H_{t-1}, E_t\right] \mathbb{P}[E_t | H_{t-1}]$$

$$\geq \left( \mathbb{P}\left[M'^{\mathcal{O}_{t-1}} \to 1 | y_{\leq t-1} = a(H_{t-1})\right] \right) \left(1 - \frac{1}{50T}\right)^{T-t} \left(1 - \frac{1}{50T}\right)$$

$$= \mathbb{P}\left[M'^{\mathcal{O}_{t-1}} \to 1 | y_{\leq t-1} = a(H_{t-1})\right] \left(1 - \frac{1}{50T}\right)^{T-(t-1)}$$

Thus by induction, (2), and (1) we conclude that

$$\mathbb{P}[V \to 1] \geq \mathbb{P}\left[M'^{\mathcal{O}_1} \to 1\right]\left(1 - \frac{1}{50T}\right)^T \geq \frac{49}{50}\left(\mathbb{P}[M^{\mathcal{O}} \to 1] - \frac{1}{150}\right)\left(1 - \frac{1}{50}\right)$$

$$\geq \left(\frac{2}{3} - \frac{1}{150}\right) \cdot \left(\frac{49}{50}\right)^2$$

$$> \frac{3}{5}$$

**Soundness**  If $x \notin L$, we first describe how the prover $B$ can efficiently follow the prescribed protocol. In each round $t$ the prover $B$ draws $R = 48d^2 \log 100T$ samples from $y_t$ conditioned on $y_{I(t)} = a_{I(t)}$ (this may involve querying $\mathcal{O}(a_{I(t)})$ up to $R$ times). Next $B$ computes the sample mean $\hat{q}_t$ of these $R$ samples. $B$ aborts if the value $\hat{p}_t$ output by $A'$ in Step 2.a does not satisfy $|\hat{q}_t - \hat{p}_t| < \frac{1}{2d}$. Finally, when $B$ is queried for a random integer in step 2.b, $B$ outputs a number $z_t^B \in \{0, \ldots, d\}$ sampled uniformly at random.

Next we analyze the probability that the verifier $V$ accepts when $B$ follows the protocol as described above, and $A'$ is an arbitrary strategy. Let $p_t = \mathbb{P}[y_t = 1 \mid y_{I(t)} = a_{I(t)}]$ be the true probability that $y_t = 1$ conditioned on the execution so far. Let $H_t$ be the history of all messages sent in the protocol up until the end of round $t$. Let $a(H_t)$ denote the bits $a_1 \ldots a_t$ output in the history $H_t$. Throughout the proof we will consider three possible events based on the behavior of $A'$ in each step. Let $E_t^0$ be the event that $|\hat{p}_t - p_t| < \frac{1}{4d}$, let $E_t^1$ be the event that $\frac{1}{4d} \leq |\hat{p}_t - p_t| < \frac{3}{4d}$, and let $E_t^2$ be the event that $|\hat{p}_t - p_t| \geq \frac{3}{4d}$. We will call a history $H_t$ "good" if $E_{t'}^0 \cup E_{t'}^1$ holds and $B$ does not abort in round $t'$ for all $t' \leq t$ in the history. Let $K_t$ be the event that $B$ aborts in round $t$.

For the analysis it will be useful to define an alternative oracle machine $M'$. The machine $M'$ is exactly the same as $M$ except that in the final step $T$, if $M$ outputs 0, then with probability $\frac{1}{25}$, $M'$ outputs 1, otherwise $M'$ outputs the same value that $M$ outputs. This implies that, given any initial setting of the transcript $y_{\leq t} = a_{\leq t}$ and any oracle $\mathcal{O}'$,

$$\mathbb{P}\left[M'^{\mathcal{O}'} \to 1 | y_{\leq t} = a_{\leq t}\right] = \mathbb{P}\left[M^{\mathcal{O}'} \to 1 | y_{\leq t} = a_{\leq t}\right] + \frac{1}{25} \cdot \mathbb{P}\left[M^{\mathcal{O}'}\right] \to 0 | y_{\leq t} = a_{\leq t}] \geq \frac{1}{25}. \tag{6}$$

The proof proceeds by induction for decreasing values of $t \leq T$. The inductive hypothesis is: For any good history $H_t$ there exists an oracle $\mathcal{O}_t$ with $\|\mathcal{O}_t - \mathcal{O}\|_\infty < \frac{1}{d}$ satisfying

$$\mathbb{P}[V \to 1 | H_t] \leq \mathbb{P}\left[M'^{\mathcal{O}_t} \to 1 | y_{\leq t} = a(H_t)\right] + \frac{T - t}{50T}. \tag{7}$$

The base case $t = T$ follows from the fact that, given a full good history $H_T$, $V$ simply outputs $a_T$. Thus, if $a_T = 1$ both $M'$ and $V$ output 1, and if $a_T = 0$, $V$ outputs 0 while $M'$ outputs 1 with probability $\frac{1}{25}$.

For the inductive step we consider three cases, resulting from conditioning on each of the $E_t^i$ for $i = \{0, 1, 2\}$.

**Conditioning on $E_t^2$.**  Observe that given any good history $H_{t-1}$, if $E_t^2$ holds then the probability that $B$ fails to abort is, again by the Chernoff bound,

$$\mathbb{P}[\overline{K}_t | H_{t-1}, E_t^2] = \mathbb{P}\left[|\hat{q}_t - \hat{p}_t| > \frac{1}{2d} \Big| H_{t-1}, E_t^2\right] \leq \mathbb{P}\left[|\hat{q}_t - p_t| > \frac{1}{4d} \Big| H_{t-1}, E_t^2\right] < \frac{1}{50T}$$

Next if $E_t^2$ holds and $B$ does abort, the probability that $V$ outputs 1 after taking $r = 192d^2 \log 100$ samples is, by the Chernoff bound,

$$\mathbb{P}[V \to 1 | H_{t-1}, E_t^2, K_t] \leq \mathbb{P}\left[|\hat{p}_t^{\mathcal{O}} - p_t| \geq \frac{3}{8d} \Big| H_{t-1}, E_t^2, K_t\right] < \frac{1}{50}.$$

Therefore, combining the two above inequalities yields,

$$\mathbb{P}[V \to 1 | H_{t-1}, E_t^2] \leq \left(1 - \frac{1}{50T}\right)\frac{1}{50} + \frac{1}{50T} < \frac{1}{25}. \tag{8}$$

**Conditioning on $E_t^0$.** Next we consider the case where $E_t^0$ occurs. $B$'s decision to abort at round $t$ depends only on the value of $\hat{p}_t$ and $\hat{q}_t$. Thus for any bit $\alpha_t$, the Chernoff bound implies that the probability that $B$ aborts is at most

$$\mathbb{P}[K_t | H_{t-1}, E_t^0, a_t = \alpha_t] \leq \mathbb{P}\left[|\hat{q}_t - p_t| > \frac{1}{4d} \Big| H_{t-1}, E_t^0\right] < \frac{1}{50T}. \tag{9}$$

Next let $H_t^1$ be the extension of $H_{t-1}$ where $E_t^0$ occurs, $a_t = 1$ and $B$ does not abort. Similarly let $H_t^0$ be the extension of $H_{t-1}$ where $E_t^0$ occurs, $a_t = 0$ and $B$ does not abort. Observe that since $B$ samples $z_t^B$ independently of everything else in the protocol, the value of $z_t = z_t^A + z_t^B \pmod 1$ in step 2.c will be uniformly random in $[0, 1]$. Thus, $a_t$ will be set to 1 with probability exactly $\hat{p}_t$ in step $t$ i.e. $\mathbb{P}[a_t = 1 | H_{t-1}, E_t^0] = \hat{p}_t$. For a good history $H_t$, let $\mathcal{O}_t$ be the oracle guaranteed to exist by the inductive hypothesis, and define $\mathcal{O}_{t-1}$ to be identical to $\mathcal{O}_t$, except that $\mathbb{P}[\mathcal{O}_{t-1}(a_{I(t)}) = 1] = \hat{p}_t$. Observe that, for any good history $H_t$, the occurrence of $E_t^0$ implies that the oracle $\mathcal{O}_{t-1}$ will satisfy $\|\mathcal{O}_{t-1} - \mathcal{O}\|_\infty < \frac{1}{d}$. Therefore, applying (9) followed by the inductive hypothesis (7) yields

$$\mathbb{P}[V \to 1 | H_{t-1}, E_t^0] \leq \left(\mathbb{P}[V \to 1 | H_t^1]\hat{p}_t + \mathbb{P}[V \to 1 | H_t^0](1 - \hat{p}_t)\right)\left(1 - \frac{1}{50T}\right) + \frac{1}{50T}$$

$$\leq \left(\left(\mathbb{P}\left[M'^{\mathcal{O}_t} \to 1 | y_{\leq t} = a(H_t^1)\right] + \frac{T-t}{50T}\right) \cdot \hat{p}_t\right.$$

$$\left. + \left(\mathbb{P}\left[M'^{\mathcal{O}_t} \to 1 | y_{\leq t} = a(H_t^0)\right] + \frac{T-t}{50T}\right) \cdot (1 - \hat{p}_t)\right)\left(1 - \frac{1}{50T}\right) + \frac{1}{50T}$$

$$= \left(\mathbb{P}\left[M'^{\mathcal{O}_{t-1}} \to 1 | y_{\leq t-1} = a(H_{t-1})\right] + \frac{T-t}{50T}\right)\left(1 - \frac{1}{50T}\right) + \frac{1}{50T}$$

$$\leq \mathbb{P}\left[M'^{\mathcal{O}_{t-1}} \to 1 | y_{\leq t-1} = a(H_{t-1})\right] + \frac{T-(t-1)}{50T}. \tag{10}$$

**Conditioning on $E_t^1$.** First observe that if $E_t^1$ occurs and $B$ aborts, since $V$ takes $r = 192d^2 \log 100$ samples, the Chernoff bound implies that,

$$\mathbb{P}[V \to 1 | H_{t-1}, E_t^1, K_t] = \mathbb{P}\left[|\hat{p}_t^\mathcal{O} - p_t| > \frac{1}{8d}\right] < \frac{1}{50} \tag{11}$$

Therefore, by (11) we have,

$$\mathbb{P}[V \to 1 | H_{t-1}, E_t^1] < \mathbb{P}\left[V \to 1 | H_{t-1}, E_t^1, \overline{K_t}\right]\mathbb{P}\left[\overline{K_t} | H_{t-1}, E_t^1\right] + \frac{1}{50}\mathbb{P}\left[K_t | H_{t-1}, E_t^1\right]$$

$$\leq \max\left\{\mathbb{P}[V \to 1 | H_{t-1}, E_t^1, K_t], \frac{1}{50}\right\} \tag{12}$$

Next let $G_t^1$ be the extension of $H_{t-1}$ where $E_t^1$ occurs, $a_t = 1$ and $B$ does not abort. Similarly let $G_t^0$ be the extension of $H_{t-1}$ where $E_t^0$ occurs, $a_t = 0$ and $B$ does not abort. As before we know that the steps 2.b -2.d guarantee that $\mathbb{P}[a_t = 1 | H_{t-1}, E_t^1] = \hat{p}_t$. Again for a good history $H_t$, let $\mathcal{O}_t$ be the oracle guaranteed to exist by the inductive hypothesis, and define $\mathcal{O}'_{t-1}$ to be identical to $\mathcal{O}_t$, except that $\mathbb{P}[\mathcal{O}'_{t-1}(a_{I(t)}) = 1] = \hat{p}_t$. Observe that, for any good history $H_t$, the occurrence of $E_t^1$ implies that the oracle $\mathcal{O}'_{t-1}$ will satisfy $\|\mathcal{O}'_{t-1} - \mathcal{O}\|_\infty < \frac{1}{d}$.

Continuing, the inductive hypothesis (7) implies that

$$\mathbb{P}\left[V \to 1 | H_{t-1}, E_t^1, K_t\right] = \mathbb{P}\left[V \to 1 | G_t^1\right]\hat{p}_t + \mathbb{P}\left[V \to 1 | G_t^0\right](1 - \hat{p}_t)$$

$$\leq \left(\mathbb{P}\left[M'^{\mathcal{O}_t} \to 1 | y_{\leq t} = a(G_t^1)\right] + \frac{T-t}{50T}\right)\hat{p}_t$$

$$+ \left(\mathbb{P}\left[M'^{\mathcal{O}_t} \to 1 | y_{\leq t} = a(G_t^0)\right] + \frac{T-t}{50T}\right)(1 - \hat{p}_t)$$

$$= \mathbb{P}\left[M'^{\mathcal{O}'_{t-1}} \to 1 | y_{\leq t} = a(H_{t-1})\right] + \frac{T-t}{50T} \tag{13}$$

Therefore combining (12) and (13) we conclude that

$$
\mathbb{P}\left[V \to 1 \mid H_{t-1}, E_t^1\right] \leq \max\left\{ \mathbb{P}\left[M'^{\mathcal{O}'_{t-1}} \to 1 \mid y_{\leq t} = a(H_{t-1})\right] + \frac{T-t}{50T}, \frac{1}{50} \right\}
$$

$$
= \mathbb{P}\left[M'^{\mathcal{O}'_{t-1}} \to 1 \mid y_{\leq t} = a(H_{t-1})\right] + \frac{T-t}{50T}
$$

$$
< \mathbb{P}\left[M'^{\mathcal{O}'_{t-1}} \to 1 \mid y_{\leq t} = a(H_{t-1})\right] + \frac{T-(t-1)}{50T} \tag{14}
$$

where the penultimate equality follows from (6).

**Putting it all together.** By (6) combined with (8), (10), and (14) we have

$$
\mathbb{P}\left[V \to 1 \mid H_{t-1}\right] = \sum_{i=0}^{2} \mathbb{P}\left[V \to 1 \mid H_{t-1}, E_t^i\right] \mathbb{P}\left[E_t^i \mid H_{t-1}\right]
$$

$$
\leq \max_{i \in \{0,1,2\}} \mathbb{P}\left[V \to 1 \mid H_{t-1}, E_t^i\right]
$$

$$
\leq \mathbb{P}\left[M'^{\mathcal{O}_{t-1}} \to 1 \mid y_{\leq t-1} = a(H_{t-1})\right] + \frac{T-(t-1)}{50T}.
$$

Here the oracle $\mathcal{O}_{t-1}$ may either come from the case where $E_t^1$ achieves the maximum or where $E_t^0$ does. Either way, $\mathcal{O}_{t-1}$ satisfies $\|\mathcal{O}_{t-1} - \mathcal{O}\|_\infty < \frac{1}{d}$ as required. Thus by induction, (6), and (1),

$$
\mathbb{P}[V \to 1] = \mathbb{P}\left[M'^{\mathcal{O}_1} \to 1\right] + \frac{1}{50}
$$

$$
= \mathbb{P}\left[M^{\mathcal{O}_1} \to 1\right] + \frac{1}{25}\mathbb{P}\left[M^{\mathcal{O}_1} \to 0\right] + \frac{1}{50}
$$

$$
\leq \frac{1}{3} + \frac{1}{150} + \frac{1}{25} + \frac{1}{50} = \frac{2}{5}.
$$

**Efficiency** Both honest provers $A$ and $B$ need to sample from $\mathcal{O}$ at most $R = O(d^2 \log T) = O(K^2 \log T)$ times for each of the $T$ steps of the machine $M$. The current head position of $M$ can be encoded in $O(\log T)$ bits, so simulating one step of $M$ requires at most $O(\log(T))$ time. $V$ needs to read the $O(l)$ relevant coordinates of $a_{I(t)}$, the locations of which can each be encoded in $O(\log T)$ bits, yielding a total of $O(l \log T)$ bits read. $V$ must further sample at most $O(d^2) = O(K^2)$ times from $\mathcal{O}$.

$\square$

# E  MISSING PROOFS FOR DEBATE WITH A WITNESS

This section gives the missing proofs for the power of debate with a witness.

*Proof of Theorem 7.1.*

**Completeness** If $x \in L$ then an honest prover $A$ can output a valid witness $w$ (i.e. satisfying $M(x, w) = 1$) and run the protocol of Figure 4 with input $(x, w)$ and machine $M$. By the completeness case of Theorem 5.3, the verifier will output 1 no matter the behavior of a potentially dishonest prover $B'$.

**Soundness** Suppose $x \notin L$. Let $w$ be any witness produced by a dishonest prover $A'$. Clearly $M(x, w) = 0$, so by the soundness case of Theorem 5.3 the verifier will always output 0.

**Efficiency** The only cost is running the protocol of Figure 4 and so the prover and verifier time are the same as Theorem 5.3. $\square$

*Proof of Theorem 7.2.*

**Completeness**    If $x \in L$ then an honest prover $A$ can output a valid witness $w$ (i.e. satisfying $M(x, w) = 1$ with probability at least $\frac{2}{3}$) and run the protocol of Figure 1 with input $(x, w)$ and machine $M$. By the completeness case of Theorem 6.2, the verifier will output 1 with probability at least $\frac{3}{5}$, no matter the behavior of a potentially dishonest prover $B'$.

**Soundness**    Suppose $x \notin L$. Let $w$ be any witness produced by a dishonest prover $A'$. Clearly $M(x, w) = 0$ with probability at least $\frac{2}{3}$. Thus, by the soundness case of Theorem 6.2 the verifier will output 1 with probability at most $\frac{2}{5}$.

**Efficiency**    The only cost is running the protocol of Figure 1 and so the prover and verifier time are the same as Theorem 6.2. □

