# OpenReview forum: "Scalabale AI Safety via Doubly-Efficient Debate"
_ICLR.cc/2024/Conference — Submitted to ICLR 2024_

### Official Review · Reviewer_a5ph · 2023-10-30

**Soundness:** 3 good
**Presentation:** 3 good
**Contribution:** 3 good
**Rating:** 6
**Confidence:** 2

**Summary:**

The paper introduces a model called the doubly-efficient debate. Here, two competing provers attempt to convince a verifier of a computation's correctness, relying on black-box judgements (like human feedback). This method ensures that any polynomial-time computation can be confirmed with just a few queries to the human judgment black-box. The model promises efficient verification, but it requires the AI models to produce comprehensive reasoning traces that can stand up to rigorous human analysis.

The researchers formalize a scenario where two AI models compete to convince a verifier, who can consult human judgment, of a solution's accuracy. The aim is to ensure the right solution is reached without excess computation and that human queries remain limited regardless of the complexity of the task. Protocols developed demonstrate success in various settings: deterministic human judgment, stochastic human judgment, and settings with witnesses.

**Strengths:**

The introduction of the "doubly-efficient debate" model is an innovative way to approach the challenge of training and oversight in AI, particularly for Large Language Models (LLMs). By pitting two models against each other to verify the correctness of their outputs, the paper seeks to streamline and make efficient the process of verification, which is a unique proposition.

The paper emphasizes real-world issues, such as the potential high-stakes consequences of language models used for drafting laws or contracts. This grounding in practical applications elevates its relevance and appeal to practitioners in the field.

The paper explores various scenarios, including deterministic and stochastic human judgment. This comprehensive approach ensures that the proposed models and protocols are tested under diverse conditions, enhancing their reliability.

**Weaknesses:**

The paper poses questions about how the model would deal with errors from the oracle, either due to incorrect judgments or stochastic nature. The mere existence of these questions signifies a lack of clarity or solution in the paper about handling such errors effectively.
Models where the oracle might make errors, either randomly or on a limited set of queries, introduce an element of unpredictability into the verification process. The paper does not seem to offer robust strategies to mitigate or address these potential errors.

Although the paper introduces a theoretical model, the practical implementation of such a model and its real-world viability are not deeply explored. The proposed model's scalability, robustness, and efficiency in real-world applications remain an open question.

**Questions:**

Could the authors clarify the connection between their protocols and the framework introduced in [1]?
[1] Du Y, Li S, Torralba A, et al. Improving Factuality and Reasoning in Language Models through Multiagent Debate[J]. arXiv preprint arXiv:2305.14325, 2023.

---

> ### Author Response · Authors · 2023-11-21
>
> We are very happy to hear that you find that our approach offers a unique proposition that would be of relevance and appeal to practitioners in the field! Thank you for your encouraging words!
>
> >"Could the authors clarify the connection between their protocols and the framework introduced in [1]? [1] Du Y, Li S, Torralba A, et al. Improving Factuality and Reasoning in Language Models through Multiagent Debate[J]."
>
> The biggest distinction is that the method of [1] is test-time only invervention. That is, several models provide answers to a question, then are prompted to re-evaluate their answers based on the output of the other models and so on until a final answer is output.
>
> In contrast, our paper is a training-time intervention. That is, we train the models via reinforcement learning to answer questions by winning debates against opposing models. So in our case the debate happens at training time and the goal is to produce models that give honest, correct, and verifiable answers.
>
> Additionally, our paper proves formal theoretical guarantees showing that our debate training method needs only a fixed constant amount of human feedback per question, even when the length of the human-verifiable argument for the correct answer is very long. This is the main high-level takeaway message of our theorems, and it has no counterpart in [1], which is an empirical, test-time method.

---

### Official Review · Reviewer_5QN9 · 2023-10-31

**Soundness:** 3 good
**Presentation:** 3 good
**Contribution:** 3 good
**Rating:** 8
**Confidence:** 2

**Summary:**

This manuscript contributes to the 'scalable oversight' literature, presenting results for an environment in which two adversarial provers argue in favor of and against a result, for human review.

**Strengths:**

**originality**

The paper seems to make improvements to an existing literature.

**quality**

The work seems to be of good quality.

**clarity**

The paper is well written and clear.

**significance**

AI models are becoming larger and more capable, making AI safety an increasingly important topic.  It seems to me that this approach - adversarial provers and human oversight - is promising.

**Weaknesses:**

Caveat: I've given myself a low confidence score as this literature is not one that I know or have worked in.  Thus, I would have benefitted from a very simple running example through the paper.  I understand that space is tight, and expect that readers actually working in this area would benefit from that less than I would, so would certainly not make including one a strong recommendation.

My main concern about this approach is that it relies on unsound reasoners, overseen by an unsound human.  While I agree that, ultimately, there are turtles or elephants all the way down, we can choose how to position the elephants/turtles.  The autoformalization project (e.g. Jiang et al.'s Draft, Sketch, Prove) relies to a greater extent on sound reasoners.  While I think that each of these approaches has distinct strengths and weaknesses, I think that they should at least be compared.

Minor typos:
1. "makes progress both" -> "makes progress on both"
1. "currently know for delegating" -> "currently known for delegating"
1. the final sentence on p.3 ("For a real number $p$...") is a fragment
1. "for it's correctness" -> "for its correctness"

**Questions:**

Can you present a simple example of a false proof that survives the protocol, because of inconsistencies or errors in the (human) oracle?  An ideal example, from my point of view, would display a subtle oracular error (e.g. a minor, buried assumption on real numbers) that spirals into a clearly false result.  Perhaps an easy way to do this would be to show the $(A, B)$ machines yielding both True and False, due to the oracle's replies.

The probabilities in Definition 3.2 stood out: are these merely illustrative (so that any result could be replaced by arbitrary constants  $a$ and $b$), or would even qualitative results derived be overturned by use of different fractions (e.g. are there critical values for these numbers)?

Definition 6.1: can you provide an example of an oracle that is not $K$-Lipschitz?

**FYI**

Not questions re: the reviewing of this paper, but the sort of questions I would ask if talking to the authors about the research more generally.
 Thus, these do _not_ need to be answered here:
1. is the argumentation procedure in Dewatripont and Tirole's "Advocates" a special case of this framework in any way?  Their approach to efficiency is different from that taken here, but may be complementary?
1. the economic theory literature also contains models of 'cheap talk' and 'long cheap debate' (like a 'debate' in the current paper), in which two biased but informed advisors make comments to a decision maker, who tries to determine the true state of the world from their comments.  In the canonical version, the comments are intervals, rather than the present probabilities.
1. Foster & Vohra's chapter on calibration, in which a decision maker attempts to identify whether or not a purported expert has true expertise, by means of repeated questioning, also seems generally related.
1. Dung's abstract argumentation framework also came to mind, explicitly considering arguments and their attacks/refutations.

---

> ### Author Response · Authors · 2023-11-21
>
> We are very encouraged by your comment that our proposed combination of  adversarial provers and human oversight is a promising direction in an increasingly important topic. We hope that our proposal can accelerate much needed progress on this subject.
>
> >"Can you present a simple example of a false proof that survives the protocol, because of inconsistencies or errors in the (human) oracle?"
>
> Please see our response below regarding the example of the oracle machine $M$ that computes majority. This example shows that if we are attempting to find the majority vote of $n$ questions where the correct probability for yes for each is $\frac{1}{2} - O(\frac{1}{\sqrt{n}})$, an error of just $O(\frac{1}{\sqrt{n}})$ in the human oracle is enough to flip the answer from 0 to 1. Thus, using debate to conduct polling on a very evenly contested issue can result in large error, even if each individual answer is only slightly biased.
>
> An important point to make here is that after training with debate, one can easily verify at test time if an oracle machine $M$ is non-Lipschitz at $\mathcal{O}$ by simply running the protocol by asking $A$ to argue that $M^{\mathcal{O}}(x) = 1$ and then running it again while asking for $A$ to show that $M^{\mathcal{O}}(x) = 0$. If $A$ is able to win the debate in both cases, then it is clear that $M$ is not sufficiently Lipschitz and the results cannot be trusted.
>
> >"The probabilities in Definition 3.2 stood out: are these merely illustrative (so that any result could be replaced by arbitrary constants
>  $a$ and $b$
> ), or would even qualitative results derived be overturned by use of different fractions (e.g. are there critical values for these numbers)?"
>
> The probabilities in Definition 3.2 are just illustrative constants, all that is needed is that they are bounded away from $1/2$. A key point here is that the probability is over the internal randomness used by $M$. Therefore by simply running the machine $M$ independently $k$ times on the same input $x$ and taking the majority vote of the answers, the probability of outputting the wrong answer goes to zero at a rate exponential in $k$.
>
> >"Definition 6.1: can you provide an example of an oracle that is not
> -Lipschitz?"
>
> Let $\epsilon = \sqrt{\frac{10}{n}}$ and $\mathcal{O}$ be the oracle that outputs 1 with probability $\frac{1}{2}-\epsilon$ on every query.
> Let $M^{\mathcal{O}}$ be the machine that, on input $x$, takes the majority vote of the outputs of $\mathcal{O}$ on the $n$ strings $y^1,\dots,y^n\in\{0,1\}^l$ lexicographically following $x$. Then $M$ is not $K$-Lipschitz for any $K < O(\sqrt{n})$.
>
> To see why consider the oracle $\mathcal{O}'$ which outputs 1 with probability $\frac{1}{2} + \epsilon$. By standard concentration bounds, the probability that the majority over $n$ samples from $\mathcal{O}$ is 1 is at most $\exp(-O(\epsilon^2 n))$. On the other hand, the probability that the majority of $n$ samples from $\mathcal{O}'$ is 0 is $\exp(-O(\epsilon^2 n))$. Thus by our choice of $n = 10\frac{1}{\epsilon^2}$ a change in the probabilities of the oracle by $2\epsilon$ can result in a change in the output of $M$ by nearly 1. Thus $M$ is not $K$-Lipschitz at $\mathcal{O}$ for $K < O(\frac{1}{\epsilon}) = O(\sqrt{n})$.
>
> > "FYI"
>
> Thank you for this list of related debate/argument concepts. Your pointers contain some very interesting ideas that could be used extend or change our debate model, and to reason about debate more generally. We look forward to engaging with these ideas actively in future work. Thank you again!

---

### Official Review · Reviewer_zvu9 · 2023-11-01

**Soundness:** 3 good
**Presentation:** 4 excellent
**Contribution:** 3 good
**Rating:** 6
**Confidence:** 3

**Summary:**

This paper studies interaction protocol for the debate framework introduced by Irving et al. 2018, focusing on computational complexity aspects. More specifically, the main motivation of this work is that the original framework assumes that the honest strategy can simulate AI systems for an exponential number of steps. The paper introduces new debate-based protocols, where this  dependence is polynomial.  Inspired by interactive proofs, the debate framework is modeled as a triple $(A, B, V)$, where $A$ and $B$ are provers and $V$ is a verifier, modeled as oracle Turing machines -- oracle models a human evaluator. The paper systematically studies the problem at hand, considering both deterministic and stochastic debate protocols. The results show that the proposed protocols are complete and sound, and that the prover that argues for the correct decision and the verifier run in a polynomial number of steps. Moreover, the number of oracles quires that the verifier requires is constant.

**Strengths:**

- The paper is well written and enjoyable to read. It introduces all the relevant content important for understanding the formal framework and the results. The practical examples provided throughout the main paper clearly motivate the protocols studied in this work, and are helpful for understanding technical details.
- To my knowledge, these results are novel, and provide a different perspective on the debate framework. The protocols are relatively simple, but yield guarantees which appear to improve those from prior work.
- The paper provides rigorous analysis of the protocols, proving that they are sound and complete, and show that the protocols are efficient in relevant parameters of the problem setting. The proofs appear to be relatively simple, and at the first glance, they seem correct.
- These results would be of an interest to researchers working on alignment problems in AI, and could spark interesting discussions on the practicality of the debate based framework in large language models.

**Weaknesses:**

- The paper primarily provides a theoretical treatment of the protocols it considers. Given that there is a concrete practical scenario that motivates these protocols, and since the protocols appear to be relatively simple, it seems that the authors could have conducted an experimental evaluation akin to the one in (Irving et al. 2018), but focusing on LLMs. Experiments that compare this work to prior debate-based approaches would be useful.

- Although the work relaxes some requirements/assumption compared to prior work, more specifically (Irving et al. 2018), it doesn't fully address all the challenges related to utilizing the debate-framework in practice. For example, the protocols rely on a relative restrictive assumption that the oracle representing human judgment is correct/unbiased. That said, these limitations are clearly discussed in the conclusion section.

**Questions:**

I don't have specific clarification questions. However, it would be great if the authors could provide a discussion related to my comments about the weaknesses of their work. More specifically, some discussion on whether it would be possible to set up an experiment based on LLMs which showcases the utility of their framework.

---

> ### Author Response · Authors · 2023-11-21
>
> We are very happy to hear that you found our paper well written and enjoyable to read with practical examples provided throughout the main paper that clearly motivate the protocols studied in this work, and are helpful for understanding technical details! This type of encouragement means a lot to us.
>
> >"I don't have specific clarification questions. However, it would be great if the authors could provide a discussion related to my comments about the weaknesses of their work. More specifically, some discussion on whether it would be possible to set up an experiment based on LLMs which showcases the utility of their framework."
>
> At the scale of the examples given in our paper (e.g. writing long contracts or conducting meta-analyses), LLMs are not currently good enough to "get-off the ground" in an RL training setup where one model produces outputs and the other points out flaws. In particular, the first model will likely always have many flaws in its outputs.
>
> However, we believe that it will soon be possible to set up smaller-scale but useful experiments with LLMs utilizing this framework. The simplest version of such an experiment would be to train two models $A$ and $B$ in an adversarial RL setup where the human judge determines who wins. For example, a natural setting could be math or logical reasoning problems that require many steps of chain-of-thought to solve. Here the model $A$ is supposed to provide long chain-of-thought answers to questions, and $B$ is supposed to point out an incorrect step in the chain-of-thought. Human raters are then shown the step pointed out by $B$. If the rater judges the step to be incorrect $B$ wins, otherwise $A$ wins.
> The base models are then trained by playing against each other to win at this task. Our method applied to this task would allow the number of human ratings required to scale with the number of problems solved, rather than the number of problems times the number of steps per problem.
>
> The key properties that such an experiment testing our theory needs to have is that:
> 1. It is possible for the models to break down complex arguments into simpler, human-judgeable steps.
> 2. The arguments made are long enough that the efficiency gained by only requiring human feedback for a single step is significant.

---

> > ### Comment · Reviewer_zvu9 · 2023-11-22
> >
> > Thank you for your response, and for providing a discussion related to my comments about potential experiments.

---

### Official Review · Reviewer_TQe8 · 2023-11-04

**Soundness:** 3 good
**Presentation:** 3 good
**Contribution:** 3 good
**Rating:** 6
**Confidence:** 2

**Summary:**

The paper proposes a doubly-efficient debate, where two polynomial-time provers compete to convince a significantly more efficient verifier that they have correctly solved a computational problem that depends on black-box access to human judgments.

**Strengths:**

See question part

**Weaknesses:**

See the question part

**Questions:**

This paper proposes two efficient-debate protocols for large language model debate. The paper is out of my research domain and it is hard for me to follow the paper. It would be appreciated if the authors could answer the question below.

1. Could authors give an example to show the difference between the proposed debate method and existing works such as Irving et al. (2018)? Why the proposed method could get a better bound intuitively?

2. How to implement the method in practice to help researchers to train an LLM?

---

> ### Author Response · Authors · 2023-11-21
>
> >1. "Could authors give an example to show the difference between the proposed debate method and existing works such as Irving et al. (2018)? Why the proposed method could get a better bound intuitively?"
>
> The key difference between our approach, and that of Irving et al. (2018) is that we explicitly require that the debater model arguing for the truth can do so efficiently. In contrast, Irving et al. (2018) assume that both debaters have unlimited computational power. To see why this can make a difference, consider the example for Theorem 6.2, in which we want to use a debate between LLMs to perform a meta-analysis of scientific literature on whether caffeine improves test-taking performance. Suppose that in this example, the truth is that the literature shows that caffeine does not improve performance. The model $A$ could attempt to $p$-hack the analysis by selecting a subset of existing studies that give the desired result, and then inventing an after-the-fact reason to reject all the studies that were not contained in the selected subset.
>
> The key difference between our approach and that of Irving et al. (2018) now arises with regard to the model $B$, which is tasked with arguing for the truth (i.e. that caffeine does not improve performance). In the approach of Irving et al. $B$ is modelled as having unlimited computational resources. So $B$ would be allowed to employ brute-force strategies in order to show that $A$ has $p$-hacked. For example, $B$ could re-run the analysis on every subset of studies the same size as the subset selected by $A$, and show that the opposite conclusion is reached a large majority of the time. Note that this strategy requires $B$ to run a potentially exponentially large number of meta-analyses in order to refute just one meta-analysis run by $A$.
>
> Thus, using the approach of Irving et al. it is possible that the debater arguing for the truth may necessarily need to spend exponentially more compute in order to actually win the debate. Our approach explicitly requires the design of protocols where the debater arguing for the truth must run efficiently. In fact, the protocols we design allow the debater arguing for the truth to win by spending only $O(T\log T)$ compute time, where $T$ is the time that it would take to solve the problem without debate. To rephrase this in the context of the meta-analysis: if it is possible to correctly perform a meta-analysis via $T$ forward passes of an LLM, then our debate protocol requires only $O(T \log T)$ forward passes in total to guarantee that the debater arguing for the truth will win.
>
> >2. "How to implement the method in practice to help researchers to train an LLM?"
>
> In practice, in the simplest version of our protocol, one would train two models $A$ and $B$ via reinforcement learning to play the following zero-sum game:
>
> 1. Sample a question $Q$ from the dataset.
> 2. Prompt $A$ to give a chain-of-thought answer to $Q$.
> 3. Prompt $B$ to point out a step in $A$'s answer which is incorrect.
> 4. Show the step pointed out by $B$ to a human rater. If it was incorrect, $B$ wins, otherwise $A$ wins.

---

### Author Response · Authors · 2023-11-21

We thank all of the reviewers for their careful reading of our paper as well as for their constructive questions and comments. Moreover, we are very grateful for your support of our work!

---

### Meta-Review · Area_Chair_GaUc · 2023-12-11

**Metareview:**

Despite the high ratings, the reviewers' confidence in this paper is relatively low. To make a more objective judgment, AC thoroughly read the paper. Overall, the paper employs the Debate framework to address some misalignment issues. Debate is a common framework in the field of AI Alignment, and the authors have effectively framed its theory and methodology.

However, in terms of the framework, I fail to see significant differences from Irving et al. (2018). Experimentally, the authors have not conducted substantial experiments on any Large Language Models (LLMs). The reviewers share this concern. I have carefully read the exchange between Reviewer TQe8 and the authors, and the process described in the authors' Response 2 does not significantly differ from that in Irving et al. (2018). Furthermore, 'the protocols rely on a relatively restrictive assumption that the oracle representing human judgment is correct/unbiased,' which seems to be a limitation of this work.

In summary, this paper makes some interesting explorations and analyses based on Irving et al. (2018) but lacks sufficient innovation. AC is concerned about the practical applicability of the proposed mechanism in LLMs. One suggestion is for the authors to conduct comprehensive experiments on some open-source LLMs to demonstrate the superiority of their framework. This paper has not met the acceptance standards of ICLR, and it is hoped that the authors will further revise and enhance the persuasiveness of their framework.

**Justification For Why Not Higher Score:**

N/A

**Justification For Why Not Lower Score:**

N/A

---

### Decision · Program_Chairs · 2024-01-16

Reject